# A Cleanable Self-Assembled Nano-SiO_2_/(PTFE/PEI)*_n_*/PPS Composite Filter Medium for High-Efficiency Fine Particulate Filtration

**DOI:** 10.3390/ma14247853

**Published:** 2021-12-18

**Authors:** Yan Luo, Zhongyun Shen, Zhihao Ma, Hongfeng Chen, Xiaodong Wang, Minger Luo, Ran Wang, Jianguo Huang

**Affiliations:** 1Department of Chemistry, Zhejiang University, Hangzhou 310027, China; 0620158@zju.edu.cn; 2Shaoxing Testing Institute of Quality and Technical Supervision, Market Supervision Administration of Shaoxing Municipahty, Shaoxing 312366, China; mzh@stiq.org (Z.M.); wangxd@stiq.org (X.W.); sxzjjd@stiq.org (M.L.); 3CAM-China Productivity Center for Machinery, China Academy of Machinery Science and Technology, Beijing 100044, China; wangran@pcmi.com.cn

**Keywords:** polytetrafluoroethylene, polyphenylene sulfide, filter medium, PM2.5, layer-by-layer self-assembly

## Abstract

A silicon dioxide/polytetrafluoroethylene/polyethyleneimine/polyphenylene sulfide (SiO_2_/PTFE/PEI/PPS) composite filter medium with three-dimensional network structures was fabricated by using PPS nonwoven as the substrate which was widely employed as a cleanable filter medium. The PTFE/PEI bilayers were firstly coated on the surfaces of the PPS fibers through the layer-by-layer self-assembly technique ten times, followed by the deposition of SiO_2_ nanoparticles, yielding the SiO_2_/(PTFE/PEI)_10_/PPS composite material. The contents of the PTFE component were easily controlled by adjusting the number of self-assembled PTFE/PEI bilayers. As compared with the pure PPS nonwoven, the obtained SiO_2_/(PTFE/PEI)_10_/PPS composite material exhibits better mechanical properties and enhanced wear, oxidation and heat resistance. When employed as a filter material, the SiO_2_/(PTFE/PEI)_10_/PPS composite filter medium exhibited excellent filtration performance for fine particulate. The PM2.5 (particulate matter less than 2.5 μm) filtration efficiency reached up to 99.55%. The superior filtration efficiency possessed by the SiO_2_/(PTFE/PEI)_10_/PPS composite filter medium was due to the uniformly modified PTFE layers, which played a dual role in fine particulate filtration. On the one hand, the PTFE layers not only increase the specific surface area and pore volume of the composite filter material but also narrow the spaces between the fibers, which were conducive to forming the dust cake quickly, resulting in intercepting the fine particles more efficiently than the pure PPS filter medium. On the other hand, the PTFE layers have low surface energy, which is in favor of the detachment of dust cake during pulse-jet cleaning, showing superior reusability. Thanks to the three-dimensional network structures of the SiO_2_/(PTFE/PEI)_10_/PPS composite filter medium, the pressure drop during filtration was low.

## 1. Introduction

With the rapid development of the social economy, the industrialization processes have not only brought a lot of materials that are of value to human beings, but also brought many negative effects to the natural environment. Air pollution, as a major environmental problem, has been arousing widespread concern. In particular, one of the primary air pollutants is fine particulate matter (PM2.5), which can enter the respiratory system, causing serious health problems and even death [1,2,3]. Air filters have been widely considered as an effective way to prevent PM2.5 emissions in the past few years, especially for the recyclable or cleanable air filter media due to their low cost and low energy consumption. Recently, various materials and technologies have been made to develop different kinds of new air filter materials, such as electrospinning [4,5,6]; electrostatic precipitation [7,8,9]; corona charging [10,11]; metal–organic framework (MOF)-based filter media [12,13,14]; self-assembled nanofibers based recyclable air filters [15]; and spun-bond [16,17,18], needle-punching [19,20,21] and melt-blown [22,23,24] nonwovens. Among these, the electrospinning fibers with small diameter and dense packing exhibit excellent filtration efficiency for fine particulate matter, but the undesirable high pressure-drop and low dust holding capacity restrict their application [25]. Electrostatic precipitators not only suffer from high energy consumption but also release some toxic gases such as ozone which are harmful to our health. For corona charging filter materials, the long-term maintenance of the electret charges under high temperature and wet conditions is still challenging [26]. The nonwoven materials with three-dimensional structures are widely used as cleanable filter media for particulate matter filtration due to their good mechanical properties, high surface-to-volume ratio and dust holding capacity, low air pressure-drop and high filtration efficiency [27].

Polytetrafluoroethylene (PTFE), para-aramid (PA), polyester fibers (PET), polyphenylene sulfide (PPS) and polyimide (P84) [28,29,30,31,32] are commonly used nonwoven materials for particulate matter filtration. Among them, the PPS nonwoven has been widely applied to waste gas filtration, especially in some industrial waste gases with high humidity due to its excellent properties such as high temperature resistance, chemical corrosion resistance and hydrolysis resistance [33]. However, there are still some disadvantages of the PPS nonwoven, such as poor wear resistance, weak oxidation resistance and low specific surface area, which lead to serious limitations in practical engineering applications [34,35]. As a filter material, the PPS possesses a desirable low pressure-drop; however, the dust filtration efficiency, especially for fine particulate, is relatively low. With the introduction of ultralow emission environmental protection policy, the emission limits for fine particulate have become stricter, which has contributed to the urgent need for high filtration efficiency, long service life and low energy cost filter media. Therefore, considerable efforts have been made by researchers in order to overcome the above-mentioned drawbacks of the PPS filter media [36,37,38].

PTFE is well known for its excellent thermal stability, oxidation resistance, low surface energy, chemical resistance and good electrical insulation [39,40,41], and it has been widely used in high-temperature filters [28,42]. The PPS nonwoven modified with PTFE component is a simple and effective way to remedy the lack of low filtration efficiency and weak oxidation resistance of the PPS filter medium. For example, hydroentangled PTFE/PPS fabric filters were prepared which showed superior filtration properties compared to a single PPS filter medium [43]. There are some techniques reported to prepare the PTFE/PPS composites, such as dip-coating [44], dip-covering [45] and spray-coating [46] methods. However, the air permeability of the filter medium prepared by the dip-covering method is quite low, which means that high filtration efficiency is always accompanied by high filtration resistance [47]. The PTFE/PPS composite filter medium prepared by the dip-coating method always generates the aggregation of PTFE particles, which is not beneficial for dust holding capacity. Moreover, the negatively charged PTFE is not easily immobilized on the surface of a PPS fiber that is negatively charged as well, leading to the desquamation of the PTFE layer in the PTFE/PPS composite filter medium after a period of application. For an excellent functionalization of the filter material, there are two exceptionally critical concepts, namely the hierarchical structures and the roughness at nanoscale on the surfaces of the substrates [48,49]. The layer-by-layer (LbL) self-assembly technique is considered an effective and environmentally friendly technique for the fabrication of composite materials with designed structures and functionalities [50], and it provides a pathway for the preparation of the above-mentioned PTFE/PPS composite filter medium. Moreover, the repeated pulse-jet cleaning for the detachment of the dust will lead to rapid abrasion of the filter medium and decrease its service life. It is known that the introduction of hard nanoparticles, such as silicon dioxide, could improve the wear performance of polymers. For instance, a PPS-PTFE/SiO_2_ composite was prepared by a simple spray process and exhibited excellent wear and corrosion resistance properties [51].

In the present study, a SiO_2_/(PTFE/PEI)*_n_*/PPS composite material was prepared through the layer-by-layer (LbL) self-assembly approach, where the positively charged polyethyleneimine (PEI) layer was firstly deposited on the surface of the PPS fiber, and then the negatively charged PTFE layer was absorbed on the surface of the PEI layer through electrostatic interaction, which was followed by the deposition of SiO_2_ nanoparticles. The SiO_2_/(PTFE/PEI)*_n_*/PPS composite filter medium obtained perfectly maintained the hierarchical network structure of the initial PPS nonwoven, and the contents of the PTFE component were easily controlled by adjusting the number of the self-assembled PTFE/PEI bilayers. The obtained SiO_2_/(PTFE/PEI)_10_/PPS composite material exhibits better mechanical properties and enhanced wear, oxidation and heat resistance in comparison with the pure PPS nonwoven. When employed as a filter material, the SiO_2_/(PTFE/PEI)_10_/PPS composite filter medium exhibited excellent filtration performance for fine particulate owing to its three-dimensional structures with high specific surface area and pore volume.

## 2. Materials and Methods

### 2.1. Materials

Silicon dioxide (SiO_2_) nanoparticles and polytetrafluoroethylene (PTFE, 60 wt.% dispersion) were purchased from Aladdin (Shanghai, China). Polyethyleneimine (PEI, typical MW = 70,000, 50 wt.% aqueous solution) was bought from Shanghai Macklin Biochemical Co. Ltd. (Shanghai, China). Triethanolamine, sodium dodecylbenzene sulfonate, silane coupling agent (KH-550) and ethanol were purchased from Sinopharm Chemical Reagent Co. Ltd. (Shanghai, China). Polyphenylene sulfide (PPS, grammage: 550 g m^−2^) and PTFE membrane-coated PPS filter medium (grammage: 550 g m^−2^) were purchased from Liaoning Xinhongyuan Environmental Protection Material Co. Ltd. (Yingkou, Liaoning, China). All the chemicals were guaranteed reagents and used without further purification. The water used was purified by a Milli-Q Advantage A10 system (Millipore, Bedford, MA, USA) with a resistivity higher than 18.2 MΩ cm^−1^.

### 2.2. Preparation of the SiO_2_ Nanoparticle Dispersion

Firstly, 5.0 g SiO_2_ nanoparticles were mixed with 50.0 mL water and ultrasonically dispersed for 30 min. Then, 2.0 mL KH-550 was added into the SiO_2_ nanoparticle dispersion, followed by stirring for 30 min and ultrasonic dispersion for 30 min. At last, 1.0 mL triethanolamine and 1.0 mL sodium dodecylbenzene sulfonate were added into the dispersion, followed by stirring for 30 min and dispersing for 30 min, resulting in the SiO_2_ nanoparticle dispersion.

### 2.3. Preparation of the SiO_2_/(PTFE/PEI)_n_/PPS Composite Filter Medium

Figure 1 presents the fabrication processes of the nano-SiO_2_/(PTFE/PEI)*_n_*/PPS composite material. A piece of PPS nonwoven placed in the suction filtration was washed by ethanol firstly and dried with air flow for 15 min (Figure 1a). Then, 30.0 mL of PEI aqueous solution (5.0 g L^−1^) was added to the filter funnel, and half of it was slowly suction-filtered off, while the rest was kept for 5 min so that the PEI was adequately adsorbed on the PPS fibers. Subsequently, 30.0 mL of water was added and filtered to remove the unabsorbed PEI, resulting in the PEI/PPS composite (Figure 1b). Then, 30.0 mL of PTFE dispersion (100 g L^−1^) was added and kept for another 5 min to make it thoroughly absorbed on the surface of the PEI layer through electrostatic interaction. Afterward, 30.0 mL of water was filtered to wash away the unassembled reagent, followed by drying in a flow of air for 15 min, yielding the PTFE/PEI/PPS composite (Figure 1c). The deposition of the PTFE/PEI double layers was repeated 5 and 10 times, and the corresponding products produced were named (PTFE/PEI)_5_/PPS and (PTFE/PEI)_10_/PPS. For the synthesis of the SiO_2_/(PTFE/PEI)_n_/PPS composite (Figure 1d), SiO_2_ nanoparticles were deposited onto the surfaces of the PTFE layers by adding 30.0 mL of the pre-prepared SiO_2_ nanoparticle dispersion to the filter funnel and keeping for 15 min. At last, the as-deposited composites were baked in the dryer (Shenzhen Huaboxing Technology Co. Ltd., Shenzhen, China) at 180 °C for 5 min, resulting in SiO_2_/(PTFE/PEI)_5_/PPS and SiO_2_/(PTFE/PEI)_10_/PPS composite filter media, respectively.

### 2.4. Characterizations

The mechanical property assessment was carried out on an INSTRON3365 instrument (INSTRON, Boston, MA, USA) with a speed of 100 mm min^−1^. The air permeability test was performed by using a YG461G instrument (NBFY, Ningbo, Zhejiang, China) operating at the pressure drop of 200 Pa, the area of 20.0 cm^2^ and the nozzle diameter of 4 mm. The wear resistance analysis was done according to ISO 12947-2:1998 (standard of International Organization for Standardization) by using the YG(B)401E Martindale pilling tester (DARONG, Wenzhou, Zhejiang, China) working with a load of 700 g. The number of rubs at which specimen breakdown occurs was recorded to evaluate the wear resistance. The field emission scanning electron microscope (FE-SEM) micrographs and EDX data were acquired on a Hitachi SU-8010 instrument (acceleration voltage: 5.0 kV, HITACHI, Tokyo, Japan) or 20.0 kV with an IXRF energy-dispersive spectrometer. The specimens for FE-SEM observation were cut into 1.50 × 1.50 cm^2^ rectangular sheets and sputtered with platinum to reduce charging. The energy-dispersive X-ray spectroscopy (EDS) mapping images were obtained on an OXFORD X-MaxN50 instrument (acceleration voltage: 15.0 kV, OXFORD, Oxford, UK). The Fourier transform infrared (FT-IR) spectra were obtained on a Nicolet iS20 apparatus (Thermo Scientific, Waltham, MA, USA) in the attenuated total reflection (ATR) mode. The differential scanning calorimeter (DSC) analyses were performed on a Mettler Toledo STARe System DSC3 instrument (Mettler Toledo, Zurich, Switzerland) in the range of 40–450 °C with a heating rate of 10 °C min^−1^ under N_2_ atmosphere. The thermogravimetric (TG) analyses were conducted on a Mettler Toledo STARe System TGA2 (Mettler Toledo, Zurich, Switzerland) in the range of 40–1100 °C with a heating rate of 10 °C min^−1^ under N_2_ atmosphere. The nitrogen adsorption−desorption isotherms were measured on an Autosorb iQ instrument (bath temperature: 77 °C, Quantachrome, New York, NY, USA).

### 2.5. Dynamic Filtration Properties

The dynamic filtration properties of the filter media were evaluated on a FEMA 1 instrument (Fil T Eq, Karlsruhe, Germany), which was made according to VDI 3926:2004 (standard of Germany). Figure 2 shows the schematic diagram of the test apparatus. At the beginning of the filtration process, the dust particles generated from the dust feeder pass through the vertical dirty-gas duct at a constant velocity of 2.0 m min^−1^, followed by being captured on the surface of the filter test sample and forming the dust cake, which could intercept the fine particulate matter efficiently. With the formation of the dust cake, the pressure drop gradually increases. When the predetermined maximum pressure-drop (1000 Pa) is reached, a cleaning pulse is activated to detach the dust cake towards the dirty-gas (upstream) side in order to regenerate the filter material. The test dust in the study was Pural NF, which contains 35% PM2.5, and the raw dust concentration was about 5.1 g m^−3^, which was monitored in real time through the photometric concentration monitor. The pulse-jet cleaning was performed at a compressed air pressure of 0.5 MPa and an electrical valve opening time of 60 ms. The whole test procedure was divided into four phases: conditioning, aging, stabilizing and measuring. In the conditioning phase, 30 loading cycles with pressure-drop controlled pulse-jet cleaning were performed with a cleaning set point of 1000 Pa. The first test phase was followed by the aging phase. The filter was exposed to 10,000 cleaning pulses at intervals of 5 s. Between the aging phase and the measuring phase, the stabilization phase (recovery of 10 loading cycles with pressure-drop controlled cleaning) was executed in order to stabilize the operating conditions and the test filter sample behavior. In the last measuring phase, several loading cycles with pressure-drop controlled pulse-jet cleaning were performed with a cleaning set point of 1000 Pa for 5 h. The PM2.5 was separated from clean gas through the PM2.5 cyclone separator. The flow rate of PM2.5 in the work was 0.85 m^3^ h^−1^. During the last phase of the test, a gravimetric evaluation of PM2.5 was performed for the calculation of the emission concentration of the PM2.5. Going through the entire test phase, the residual pressure-drop, cleaning cycle time and residual dust mass were analyzed to evaluate the relevant long-term operational properties (filtration and cleaning behavior) and emission of filter media.

## 3. Results and Discussion

### 3.1. Physical Properties of the Filter Media

The mechanical properties, air permeability and wear resistance are the main physical properties of the filter medium and directly influence its service life and filtration performance. As shown in Table 1, the meridional and latitudinal strength of the SiO_2_/(PTFE/PEI)_10_/PPS composite filter medium are 1425.6 N and 1753.4 N, respectively, and are higher than those of the PPS filter medium and (PTFE/PEI)_10_/PPS composite medium. The wear resistance of the SiO_2_/(PTFE/PEI)_10_/PPS filter medium is 125 times, which increased by 66.7% as compared with that of the PPS filter medium. The improvement in the mechanical properties and wear resistance of the SiO_2_/(PTFE/PEI)_10_/PPS composite filter medium indicates its service life will be prolonged. However, as compared with the PPS filter medium, the air permeability of the SiO_2_/(PTFE/PEI)_10_/PPS composite filter medium decreased due to the deposition of SiO_2_/PTFE/PEI multilayers on the surfaces of the PPS fibers, which led to a narrower space between the fibers. Thanks to the three-dimensional network structures, the air permeability of the SiO_2_/(PTFE/PEI)_10_/PPS composite filter medium was much better than that of the commercial PTFE membrane-coated PPS filter material.

### 3.2. Structural Characterizations of the Filter Media

As illustrated in Figure 1, the PTFE/PEI double layers were deposited on the surfaces of the PPS fibers through the LbL self-assembly technique, and then the SiO_2_ nanoparticles were immobilized thereon to give the SiO_2_/(PTFE/PEI)*_n_*/PPS composite filter medium. The FE-SEM images of the PPS filter medium are shown in Figure 1. Figure 1a exhibits the low-magnification FE-SEM image of the sample, showing the three-dimensional network structure which consisted of microfibers with a diameter of ca. 15 μm. The high-magnification FE-SEM image shown in Figure 1b exhibits that the surface of the PPS filter medium was smooth.

Figure 2 shows the morphologies of the SiO_2_/(PTFE/PEI)*_n_*/PPS composite filter medium with varied numbers of the PTFE/PEI bilayers produced by the self-assembly processes. The low-magnification FE-SEM images of the SiO_2_/(PTFE/PEI)_5_/PPS and SiO_2_/(PTFE/PEI)_10_/PPS composite filter media shown in Figure 2a,c indicate that the original three-dimensional network structures of the PPS filter medium were faithfully maintained. The high-magnification FE-SEM images of the SiO_2_/(PTFE/PEI)_5_/PPS and SiO_2_/(PTFE/PEI)_10_/PPS composite filter media are shown in Figure 2b,d. It was found that a part of the surface of the PPS filter fiber was naked when the number of the PTFE/PEI bilayers was five. By increasing the number of the PTFE/PEI bilayers to 10, the PPS microfibers were found to be almost completely coated with the SiO_2_/PTFE/PEI multilayers. It is noted that the SiO_2_/PTFE/PEI coating loaded on the surface of the PPS microfiber did not agglomerate, and there was no nanoparticle agglomerate between the spaces of the microfibers. The EDX analyses shown in Appendix A demonstrate that the composite filter media were composed of C, O, F, S and Si elements, and the amount of F element increased with the increase in the number of the PTFE/PEI bilayers; that is, the amount of PTFE component in the SiO_2_/(PTFE/PEI)*_n_*/PPS composite filter medium could be easily controlled by adjusting the number of the PTFE/PEI bilayers.

The elemental distributions of the SiO_2_/(PTFE/PEI)_10_/PPS composite filter medium shown in the EDS mapping micrographs (Figure 3a–f) demonstrate the presence of C, O, F, S and Si elements in the composite filter medium. They reveal that the S element was mainly centered in the core region of the microfibers, while the F and Si elements were mostly distributed in the outer edges.

Figure 4a exhibits the FT-IR spectra of the pure PPS filter medium before and after oxidation treatment for 400 h. The bands located at 1570 and 1469 cm^−1^ are attributed to the nonsymmetric phenyl ring stretching modes (C_6_H_6_–S). The bands at 1383, 1089 and 1007 cm^−1^ are assigned to the C–S stretching and bending vibration. The strong absorption band centered at 807 cm^−1^ is ascribed to the bending vibration of para-disubstituted aromatic rings [52]. Moreover, the peaks at 1178 and 1070 cm^−1^ are assigned to the stretching vibration of –SO_2_– and –SO–, respectively. The existence of –SO– and –SO_2_– is because of the low bond energy of the C–S bond, which was easily oxidized during synthesis, drying and storage [53]. After the oxidation treatment for 400 h, a new band located at 1230 cm^−1^ appeared, which is attributed to the sulfite linkage (–O–SO–O–), indicating the oxidation of the PPS filter medium [54]. The spectra of the (PTFE/PEI)_10_/PPS composite filter medium before and after oxidation treatment for 400 h are shown in Figure 4b. The bands located at 1571, 1470, 1384, 1090, 1008 and 808 cm^−1^ are attributed to the characteristic absorption of PPS, which is consistent with the pure PPS filter medium shown in Figure 4a. After the modification of PTFE, new bands located at 1207, 1152 and 639 cm^−1^ appeared, which are attributed to the –CF_2_– stretching and rocking vibrations [55,56]. This demonstrated that the PTFE was successfully self-assembled on the surfaces of the PPS fibers. It is worth noting that there is no apparent change of the absorption peak of the SiO_2_/(PTFE/PEI)_10_/PPS composite filter medium after the oxidation treatment for 400 h, indicating that the deposition of PTFE/PEI bilayers inhibits the oxidation of the PPS filter medium.

Figure 5a exhibits the DSC curves of the PPS filter medium and SiO_2_/(PTFE/PEI)_10_/PPS composite filter medium upon heating treatment at 450 °C. The endothermic peak located at 278.9 °C is assigned to the melting temperature of the PPS filter medium [33]. For the SiO_2_/(PTFE/PEI)_10_/PPS composite filter medium, two endothermic peaks located at 281.1 °C and 335.1 °C appeared, which are corresponding to the melting temperatures of PPS and PTFE, respectively. It is noted that the PPS melting temperature of the SiO_2_/(PTFE/PEI)_10_/PPS composite filter medium is higher than that of the pure PPS filter medium due to the modification of SiO_2_/PTFE/PEI multilayers on the surfaces of the PPS fibers. In addition, the PTFE melting temperature of the SiO_2_/(PTFE/PEI)_10_/PPS composite filter medium is also higher than its general melting point (327 °C) [57], indicating the PPS and PTFE had a synergy for heat resistance. The TG curves of the PPS filter medium and SiO_2_/(PTFE/PEI)_10_/PPS composite filter medium shown in Figure 5b indicate two weight loss stages. It is observed that both filter media have good thermal stabilities, and the weight losses were less than 5% at 500 °C. The PPS filter medium begins decomposition at 508.1 °C. For the SiO_2_/(PTFE/PEI)_10_/PPS composite filter medium, the initial decomposition temperature is 526.9 °C, which is much higher than that of the PPS filter medium, demonstrating the improved heat resistance of the SiO_2_/(PTFE/PEI)_10_/PPS composite filter medium.

The nitrogen adsorption–desorption isotherm of the SiO_2_/(PTFE/PEI)_10_/PPS composite filter medium is shown in Figure 6a, which is corresponding to the type IV curves with typical hysteresis loops based on the International Union of Pure and Applied Chemistry (IUPAC) classification. It was observed that the nitrogen adsorption capacity of the SiO_2_/(PTFE/PEI)_10_/PPS composite filter medium was much higher than that of the pure PPS filter medium, indicating that there are more pores in the SiO_2_/(PTFE/PEI)_10_/PPS composite filter medium. When P/P_0_ > 0.9, the amount of nitrogen adsorption of the SiO_2_/(PTFE/PEI)_10_/PPS composite filter medium increased sharply because of the capillary condensation. The existence of the H3 hysteresis loop in the SiO_2_/(PTFE/PEI)_10_/PPS composite filter medium indicates that there were a large number of slit-like and open pores [26]. The Brunauer−Emmett−Teller (BET) surface area of the SiO_2_/(PTFE/PEI)_10_/PPS composite filter medium was 24.419 m^2^ g^−1^ (Appendix A), which was greatly improved as compared with that of the pure PPS filter medium due to the rougher surface and the large specific surface area of the SiO_2_/(PTFE/PEI) multilayers deposited on the surfaces of the PPS fibers. Figure 6b exhibits that the SiO_2_/(PTFE/PEI)_10_/PPS composite filter medium has a hierarchical pore structure. The Barrett–Joyner–Halenda (BJH) adsorption summary pore volumes of the pure PPS filter medium and SiO_2_/(PTFE/PEI)_10_/PPS composite filter medium were 0.004 and 0.034 cm^3^ g^−1^, respectively, indicating an enhancement of almost 9 times compared to the SiO_2_/(PTFE/PEI)_10_/PPS composite filter medium.

### 3.3. Dynamic Filtration Characterization of the SiO_2_/(PTFE/PEI)_10_/PPS Composite Filter Medium

The dynamic filtration characterization of the SiO_2_/(PTFE/PEI)_10_/PPS composite filter medium was investigated in a filter test apparatus mentioned in Section 2.5 by using the Pural NF as the test dust. At the beginning of the filtration process, the dust particles are captured on the surface of the filter and form the dust cake which prevents the dust emission [58]. Figure 7 shows the superimposed representation of the pressure-drop curves versus time of selected loading cycles of the PPS filter medium and SiO_2_/(PTFE/PEI)_10_/PPS composite filter medium. It indicates that the pressure drop of both filter media increases gradually with time due to the formation of the dust cake. Because of the smooth surface of the PPS filter medium, the dusts direct penetrate through the filter material more easily than through the SiO_2_/(PTFE/PEI)_10_/PPS composite filter medium during the periods when the medium surface was not protected by the dust cake, resulting in low pressure-drop.

In order to regenerate the filter medium, the dust cake was frequently detached through the reverse-flow pulse-jet cleaning. However, the particles which had penetrated into the porous medium and the occurrence of patchy cleaning would lead to the existence of the residual dust layer; as a consequence, a pressure drop happened after cleaning. The pressure drop recorded across the filter medium shortly after the pulse-jet cleaning is called residual pressure-drop. It is desirable to have a low and stable residual pressure-drop during the filtration process. Figure 8a exhibits the development of residual pressure-drop versus time before and after aging of the PPS filter medium and SiO_2_/(PTFE/PEI)_10_/PPS composite filter medium. The initial residual pressure-drop of the SiO_2_/(PTFE/PEI)_10_/PPS composite filter medium was higher than that of the PPS filter medium due to the deposition of SiO_2_/PTFE/PEI multilayers, which led to the decrease in air permeability [59]. It was reported that the cleaning efficiency decreased with the increase in the number of cleaning cycles [60]. The residual pressure-drop of the two filter materials increases with the cycle in the conditioning phase (prior to aging). Therein, the residual pressure-drop of the SiO_2_/(PTFE/PEI)_10_/PPS composite filter medium was higher than that of the PPS filter medium. This is because the SiO_2_/PTFE/PEI multilayers immobilized on the surfaces of the PPS fibers increase the specific surface area and pore volume, resulting in more dust being absorbed on the surfaces of the fibers, which led to the increase in the residual pressure-drop of the SiO_2_/(PTFE/PEI)_10_/PPS composite filter medium. It is noted that the residual pressure-drop of the SiO_2_/(PTFE/PEI)_10_/PPS composite filter medium prepared in this work was much lower than that of the reported PTFE/PPS composite filter medium due to the three-dimensional network structure inherited from the PPS [43]. After aging, the residual pressure-drop of the SiO_2_/(PTFE/PEI)_10_/PPS composite filter medium rises much slower as compared with that of the PPS filter medium, and the residual pressure-drops of both are close to each other. This is because the PTFE layers coated on the surfaces of the PPS fibers possess low surface energy, which reduces the adhesion between the dust cake and filter material fiber and makes it easier for the dust to peel off [61].

Cleaning cycle time refers to the time interval between two reverse-flow pulse-jet cleanings of the filter material. The cleaning cycle time has an important influence on the lifetime of the filter material. The short cleaning cycle time means frequent cleaning, making the filter bag more vulnerable to being damaged. Figure 8b reveals that the cleaning cycle time of the PPS filter medium decreases rapidly, especially prior to aging. For the SiO_2_/(PTFE/PEI)_10_/PPS composite filter medium, the cleaning cycle time decreases more slowly. After aging, the cleaning cycle time of the SiO_2_/(PTFE/PEI)_10_/PPS composite filter medium almost remains unchanged and is close to that of the PPS filter medium. This result indicates that the modification of PPS fibers with the SiO_2_/PTFE/PEI multilayers improves the stability of the modified PPS filter medium.

The filtration performances of different filter media were evaluated after aging. The PM2.5 emission concentration (*C*_*PM*2.5_) was calculated using the following Equation (1):
(1)$$C_{\mathrm{PM}2.5}=\frac{m-m_{0}}{Q\times t},$$
where the *m*_0_, *m*, *Q* and *t* correspond to the mass of the absolute filter before measuring, the mass of the absolute filter after measuring, the flow rate of PM2.5 and the measuring time, respectively. The filtration efficiency of PM2.5 (*η*) was calculated using the following Equation (2):
(2)$$\eta =\frac{C_{\mathrm{dust}}\times 0.35-C_{\mathrm{PM}2.5}}{C_{\mathrm{dust}}\times 0.35}\times 100\%$$
where *C_dust_* corresponds to the raw dust concentration. The results are listed in Table 2 and reveal that the PPS filter medium and SiO_2_/(PTFE/PEI)_10_/PPS composite filter medium have PM2.5 emission concentrations of 0.096 and 0.008 g m^−3^ and PM2.5 filtration efficiencies of 94.62% and 99.55%, respectively. This suggests that the filtration performance of the SiO_2_/(PTFE/PEI)_10_/PPS composite filter medium is much better than that of the PPS filter medium. It was reported that the dust cake plays a predominant role in the filtration effect of fine particles. The dust emission concentrations of the pulse-jet cleaned filter materials were measured only during the periods following every cleaning pulse due to the absence of the dust cake, indicating an inconsecutive emission pattern [62]. It has been demonstrated that the SiO_2_/PTFE/PEI multilayers self-assembled on the surfaces of the PPS fibers increase the specific surface area and pore volume of the filter material, which is beneficial to the capture of the dust particles. Hence, the formation of the dust cake on the surface of the SiO_2_/(PTFE/PEI)_10_/PPS composite filter medium occurs more easily than that of the PPS filter medium, resulting in intercepting the PM2.5 more efficiently, which led to lower PM2.5 emission concentration and enhanced filtration efficiency. In addition, the SiO_2_/PTFE/PEI multilayers deposited on the surfaces of the PPS fibers reduce the spaces between the fibers, which is in favor of the entrapment of the fine particulate matter. Moreover, the SiO_2_/(PTFE/PEI)_10_/PPS composite filter medium possesses excellent reusability owing to the low surface energy of the PTFE layers, which makes the dust cake easier to clean.

The results of wear resistance of the PPS filter medium and SiO_2_/(PTFE/PEI)_10_/PPS composite filter medium after measuring are shown in Appendix A, revealing that the wear resistance of SiO_2_/(PTFE/PEI)_10_/PPS composite filter medium before and after measuring was almost unchanged, while that for the PPS filter medium decreases obviously. In addition, the TG curves of the filter media after measuring shown in Appendix A exhibit that the initial decomposition temperatures of the PPS filter medium and SiO_2_/(PTFE/PEI)_10_/PPS composite filter medium are 504.9 and 526.5 °C, respectively. The initial decomposition temperature of the SiO_2_/(PTFE/PEI)_10_/PPS composite filter medium after measuring was almost unchanged as compared with that of before measuring, while that for the PPS filter medium decreases slightly, indicating excellent thermal stability of the SiO_2_/(PTFE/PEI)_10_/PPS composite filter medium. The enhancement of the wear resistance and heat resistance of the SiO_2_/(PTFE/PEI)_10_/PPS composite filter medium can prolong the service life of the filter material.

## 4. Conclusions

In summary, a cleanable SiO_2_/(PTFE/PEI)*_n_*/PPS composite filter medium was prepared using the commercial PPS filter medium as the substrate via a layer-by-layer self-assembly approach. The SiO_2_/(PTFE/PEI)*_n_*/PPS composite filter medium maintained the three-dimensional network structures of the original PPS filter medium, and the SiO_2_/PTFE/PEI multilayers were uniformly deposited on the surfaces of the PPS fibers. The contents of the PTFE component can be conveniently regulated by altering the number of the PTFE/PEI bilayers. The obtained SiO_2_/(PTFE/PEI)_10_/PPS composite filter medium exhibits better mechanical properties and enhanced wear, oxidation and heat resistance as compared with the pure PPS filter medium. When used as the filter material, the composite filter medium showed outstanding filtration performance for fine particulate owing to the PTFE layers being beneficial not only to the formation of the dust cake, but also to the detachment of the dust cake during pulse-jet cleaning, resulting in high efficiency of fine particulate filtration and superior reusability. This strategy gives insight into the fabrication of functional filter materials with high efficiency and low resistance, which can be applied in the field of ultralow dust emission.

## Data Availability

The data presented in this study are available on request from the corresponding author.

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
