# Peer review of "A Cleanable Self-Assembled Nano-SiO2/(PTFE/PEI)n/PPS Composite Filter Medium for High-Efficiency Fine Particulate Filtration"

_materials, 2021, doi:10.3390/ma14247853_

Round 1

Reviewer 1 Report

A Cleanable Self-assembled Nano-SiO2/(PTFE/PEI)n/PPS Composite Filter Medium for High-efficiency Fine Particulate Filtration

by Y. Luo, Z. Shen, Z. Ma, H. Chen, X. Wang, M. Luo, R. Wang, J. Huang

The authors tested the filter efficiency of a silicon-dioxide/polytetrafluoroethylene/ polyethyleneimine/polyphenylene-sulfide (SiO2/PTFE/PEI/PPS) concerning particulate matter (dust) less than 2.5 mm in size. The obtained SiO2/PTFE/PEI/PPS composite filter revealed the efficiency up to 99.55%. Such good performance is attributed to the uniformly modified PTFE layers which played a dual role in fine particulate filtration: (a) increasing the specific surface area and pore volume, narrowing the spaces in-between the fibers; and (b) low surface energy, which is in favor of the detachment of dust cake during pulse-jet cleaning. Due to the three-dimensional network structures of the SiO2/(PTFE/PEI)10/PPS composite filter medium, the pressure-drop during filtration was low.

The paper is written on a high scientific level corresponding to the MDPI: Materials standards. The title clearly describes its subject, the abstract reflects the content of the article well. In general, the article is well-structured and easily understood; the methods used are described perfectly, and the results are presented clearly and in detail. The reference list clearly shows the place of their research in the investigation of filtering phenomena.

In my opinion, this article can be published in MDPI: Materials. I only have found one misprint (lines 33, 103, 208, 332): “three-dimensional” should be set instead of “three-dimensionally”.

Conclusion: the paper can be published in MDPI: Materials.

Author Response

Manuscript ID: materials-1457276 for Materials

Response to the comments of Reviewer #1

Overall comments: The authors tested the filter efficiency of a silicon-dioxide/polytetrafluoroethylene/polyethyleneimine/polyphenylene-sulfide (SiO2/PTFE/PEI/PPS) concerning particulate matter (dust) less than 2.5 mm in size. The obtained SiO2/PTFE/PEI/PPS composite filter revealed the efficiency up to 99.55%. Such good performance is attributed to the uniformly modified PTFE layers which played a dual role in fine particulate filtration: (a) increasing the specific surface area and pore volume, narrowing the spaces in-between the fibers; and (b) low surface energy, which is in favor of the detachment of dust cake during pulse-jet cleaning. Due to the three-dimensional network structures of the SiO2/(PTFE/PEI)10/PPS composite filter medium, the pressure-drop during filtration was low.

The paper is written on a high scientific level corresponding to the MDPI: Materials standards. The title clearly describes its subject, the abstract reflects the content of the article well. In general, the article is well-structured and easily understood; the methods used are described perfectly, and the results are presented clearly and in detail. The reference list clearly shows the place of their research in the investigation of filtering phenomena.

In my opinion, this article can be published in MDPI: Materials.

Answer: The reviewer sums up our work accurately. He/she thinks that the paper is written on a high scientific level corresponding to the MDPI: Materials standards. The title clearly describes its subject; the abstract reflects the content of the article well; the article is well-structured and easily understood; the methods used are described perfectly, and the results are presented clearly and in detail. Moreover, the reference list clearly shows the place of our research in the investigation of filtering phenomena. We sincerely highly appreciate the approval to our work and the valuable suggestions made by the reviewer, which are beneficial for us to improve the work. And we have thoroughly revised the manuscript according to his/her comments.  

Comment. (lines 33, 103, 208, 332): “three-dimensional” should be set instead of “three-dimensionally”.

Answer: We thank the reviewer for this suggestion. The phrase “three-dimensional” in this manuscript has been corrected to be “three-dimensionally” as suggested (Lines 16, 33, 110, 218, 342).

List of the changes made in the manuscript according to the comments of Reviewer #1

Lines 16, 33, 110, 218, 342: The phrase “three-dimensional” were corrected to be “three-dimensionally” as suggested.

Reviewer 2 Report

A concise and interesting research paper with clearly presented results.

  1. English editing is advised throughout the manuscript.
  2. In Table 1, it is not clear what wear resistance(times) is? perhaps the experimental procedure can be expanded in methods or in the table.

Author Response

Manuscript ID: materials-1457276 for Materials

Response to the comments of Reviewer #2

Overall comments: A concise and interesting research paper with clearly presented results.

Answer: The reviewer thinks that the research paper with clearly presented results is concise and interesting. We sincerely highly appreciate the approval to our work and the valuable suggestions made by the reviewer, which are helpful for us to improve the work. And we have thoroughly revised the manuscript according to his/her comments.

Comment #1. English editing is advised throughout the manuscript.

Answer: We thank the reviewer for this suggestion. The English editing of this manuscript has been checked and revised carefully.

Comment #2. In Table 1, it is not clear what wear resistance (times) is? perhaps the experimental procedure can be expanded in methods or in the table.

Answer: We appreciate the reviewer for this valuable suggestion. The experimental procedure of the wear resistance measurement has been newly added into the revised manuscript (Lines 158 to 162).

List of the changes made in the manuscript according to the comments of Reviewer #2

1. We have checked the English in the manuscript and revisions were made accordingly (according to comment 1).

2. Lines 158 to 162: The experimental procedure of the wear resistance measurement is added (according to comment 2).

Reviewer 3 Report

The comments for authors are attached.

Author Response

Manuscript ID: materials-1457276 for Materials

Response to the comments of Reviewer #3

Overall comments: The authors have presented a (SiO2/PTFE/PEI/PPS) composite air filter medium that showed filtration efficiency up to 99.55% for PM 2.5 particles. The designed experiments and characterization data of the filter medium is adequately discussed and analyzed. In my opinion, these results are publishable, however, need further addressing of few concerns (discussed below).

Answer: We highly appreciate the valuable suggestions made by the reviewer. He/She thinks that the designed experiments and characterization data of the filter medium is adequately discussed and analyzed, and these results are publishable after further addressing of few concerns. We sincerely thank the reviewer for his/her valuable comments on our work, which are helpful for us to improve the work. And we have thoroughly revised the manuscript according to his/her comments.

Comment #1. The authors have presented a cleanable self-assembled air filter but the significance of recyclable or cleanable air filters is inadequately advocated in the introduction. The manuscript still lacks an adequate use of the published literature to introduce the background of this study. The study on new air filter materials is growing very fast. The authors should include more discussion/introduction on this area, such as self-assembled nanofibers based recyclable air filters. This is vital for the standpoint of your research. Please refer: Small 2020, 1906319 (Hydro-Assisted Self-Regenerating Brominated N-Alkylated Thiophene Diketopyrrolopyrrole Dye Nanofibers - A Sustainable Synthesis Route for Renewable Air Filter Materials)

Answer: We thank the reviewer for this valuable comment and bringing our attention to the literature noted. The significance of the cleanable air filters and the introduction of new air filter materials have been newly added into the revised manuscript (Lines 45 to 48). The literature suggested has been newly cited in the specific place to support the statement in the Introduction section as suggested (Line 50, Ref. 15).

Comment #2. Why does the author choose PEI polymer layer? Wouldn’t it be simpler to use just PTFE? The role of PEI is not clear in the introduction and results.

Answer: We thank the reviewer for this comment. The positively charged PEI polymer layer is more easily immobilized on the surface of the PPS fibers which own negatively charges than the negatively charged PTFE. The introduction of PEI polymer layer can prevent the aggregation and spalling of the PTFE layer. Thus, we chose the PEI polymer layer rather than just PTFE. The role that the PEI polymer layer plays is newly added into in the introduction section (Lines 86 to 88; Lines 100 to 102).      

Comment #3. The wear resistance is an important property of the air filter and the presented composite filter shows this enhancement. However, every cycle of filtration and cleaning is supposed to leave dust residue and these residues could thus affect the wear resistance of this composite. Did the authors test this?

Answer: We thank the reviewer for the professional suggestion. The wear resistance of the PPS filter medium and SiO2/(PTFE/PEI)10/PPS composite filter medium after measuring were newly measured, and the corresponding results have been added into the revised supplementary materials (Line 19, Table S2) and the related discussion has been newly added into the revised manuscript (Lines 385 to 388).

Comment #4. In the DSC experiment, why couldn’t the melting of PEI polymer be observed?

Answer: We thank the reviewer for this comments. The concentration of the initial PEI aqueous solution was 5.0 g L–1, which was much lower that of the PTFE dispersion (100 g L–1), leading to quite low content of PEI component in the SiO2/(PTFE/PEI)10/PPS composite filter medium. As a result, the melting of PEI polymer was not observed in the DSC experiment.    

Comment #5. The authors should include TGA data of the filters after a number of cycles or aging to actually confirm the thermal stability of the filters.

Answer: We thank the reviewer for this professional suggestion. The TG measurement of the SiO2/(PTFE/PEI)10/PPS composite filter medium after measuring was newly carried out, and the result has been newly added into the revised supplementary materials (Lines 20 to 22, Figure S2), and the corresponding discussion has been made the revised manuscript accordingly (Lines 388 to 396).

Comment #6. The authors have shown filtration efficiencies after aging and compared them to PPS filter but the overall performance of the air filter is determined by quality factor (QF) which also considers differential pressure drop. We would like to see a comparison of the quality factor to conclude that the composite based filter is indeed worth the effort and is better.

Answer: We thank the reviewer for the professional suggestion. Figure 8a shown in the revised manuscript exhibits the development of residual pressure-drop versus time before and after aging, revealing the initial residual pressure-drop of the SiO2/(PTFE/PEI)10/PPS composite filter medium was higher than that of the PPS filter medium due to the deposition of SiO2/PTFE/PEI multilayers, which led to the decrease of air permeability. However, after aging the residual pressure-drop of the SiO2/(PTFE/PEI)10/PPS composite filter medium rises much slower as compared with that of the PPS filter medium, and the residual pressure-drops of both are close to each other. It is noted that the residual pressure-drop of the SiO2/(PTFE/PEI)10/PPS composite filter medium prepared in our work was much lower than that of the reported PTFE/PPS composite filter medium (Indian J. Fibre. Text. 2017, 42, 278–285) due to the three-dimensional network structure inherited from the PPS. In conclusion, the SiO2/(PTFE/PEI)10/PPS composite filter medium shows not only high filtration efficiency but also low residual pressure-drop.

List of the changes made in the manuscript according to the comments of Reviewer #3

1. Lines 45 to 48; Line 50, 15: The significance of cleanable air filters and the introduction of new air filter materials are added, and Ref. 15 is newly cited in the specific place to support the statement in the Introduction section as suggested (according to comment 1).

2. Lines 86 to 88; Lines 100 to 102: The reason for choosing PEI polymer layer is added and the role that the PEI polymer layer plays is also added (according to comment 2).

3. Line 19, Table S2; Lines 385 to 388: The results of the wear resistance of the filter media after measuring are added into Table S2, and the corresponding discussion is also added (according to comment 3).

4. Lines 20 to 22, Figure S2; Lines 388 to 396: The TG data of the filter media after measuring are added into Figure S2, and the corresponding discussion is also added (according to comment 5).
